# And in the Middle of My Chaos There Was You?—Dog Companionship and Its Impact on the Assessment of Stressful Situations

**DOI:** 10.3390/ijerph16193664

**Published:** 2019-09-29

**Authors:** Christine Krouzecky, Lisa Emmett, Armin Klaps, Jan Aden, Anastasiya Bunina, Birgit U. Stetina

**Affiliations:** 1Psychological Outpatient Clinic, Sigmund Freud University Vienna, 1020 Wien, Austria; lisa.emmett@sfu.ac.at (L.E.); armin.klaps@sfu.ac.at (A.K.); birgit.u.stetina@sfu.ac.at (B.U.S.); 2Faculty of Psychology, Sigmund Freud University Vienna, 1020 Wien, Austria; jan.aden@sfu.ac.at (J.A.); anastasiya.bunina@sfu.ac.at (A.B.)

**Keywords:** human-dog relationship, critical life events, sense of coherence, role-specific responsibilities, inner- and cross-species relationships

## Abstract

Recent studies show evidence that human-dog companionship has healthy effects on humans. For example, findings demonstrate that owning a dog leads to a reduction in stress levels. Aspects that have not been taken into consideration so far are underlying theoretical principles of stress like the sense of coherence (SOC) by Aaron Antonovsky. The SOC consists of psycho-social, biochemical and physical conditions which indicate whether or not inner and outer stimuli are comprehensive, manageable and meaningful to an individual. In addition, it is still unclear if owning a dog affects the subjective assessment of critical life events (CLE), which is associated with the strength of the SOC (the stronger the SOC, the better the handling and assessment of stressful situations). Based on these aspects, the goal of the study was to examine if dog ownership, as well as values of the SOC, have an impact on the subjective evaluation of CLE (including daily hassles as well as unexpected critical life events). For this purpose, dog owners and non-dog owners were surveyed online and were compared based on their personal estimations regarding these constructs. Statistical analysis including t-tests, correlations and interaction-analyses were performed and a significant difference between dog owners and non-dog owners regarding the assessment of daily hassles was found. Contrary to expectations, results show that dog owners assessed daily stressors to be more stressful than non-dog owners did. Moreover, data show that the higher the number of stated relationships (inner- and cross-species), the more stressful life events were assessed to be. Calculations showed no evidence for the influence of dogs regarding the SOC. Based on the actual findings, it might be assumed, that an overestimation of the dog’s protective role regarding stress has taken place in public media and in research as well.

## 1. Introduction

Over the past decades, dogs have assumed many significant roles within human society, ranging from hunting partner to companion animal [1]. Nowadays, dogs are increasingly important within the family system [2] and they often gain a special role as social partners. These aspects are accompanied by a growing interest in the field of the human-dog relationship.

Most of the research in this area is focused on bio-psycho-social effects of the human–animal bond in general. Overall, most studies concerning this topic report positive influences of animals on the humans’ well-being. In this context, positive effects of dogs are not only found in short-term interactions with therapy dogs but also within long-term interactions like dog ownership [3]. In addition to findings regarding lower blood pressure, as well as lower cholesterol levels, a long-term study indicates that dog owners have a significantly lower probability to suffer from cardiovascular diseases [4]. Furthermore, there is evidence of reduced cortisol levels, as well as an increase of oxytocin during human–animal interactions, which are both hormones that play an important role regarding stress regulation [5]. Moreover, data also suggests positive effects of pet ownership concerning social–emotional competences like affection and emotional warmth [6]. Despite these findings, which underline the positive influence of dogs on human health, it has to be mentioned that there are also contrary results which rarely receive attention. Harold Herzog (2011) [7] emphasized that the “generalized pet effect” on human health and happiness remains a hypothesis because of inconsistent results. The author’s explanation of the “pet effect” (the overly positive interpretation of existing results) is that so far too little attention has been paid to the critical aspects of the human–animal relationship [7].

Although a lot of research is focused on the effects of dog companionship, there is still a lack of knowledge about how the relationship with the animal influences humans during stressful times. Dealing with this topic seems highly relevant, since people are regularly confronted with situations that disturb their daily routine and entail a reorientation [8].

Stressful situations include daily hassles (normative stressors), as well as critical life events (non-normative stressors). Overcoming these circumstances is connected to unique challenges which usually cause specific stress-responses [8]. Healthy humans usually draw several strategies from the large amount of resources available to cope with stress and restore their skills that are necessary for everyday life [9]. Studies concerning this topic demonstrate that individuals differ in terms of their available resources, as well as in their abilities to cope with stressful situations [10]. Aaron Antonovsky traced this phenomenon back to certain salutogenetic conditions (psycho-social, biochemical and physical), which determine how stressors can be successfully managed with the available resources [10]. Antonovsky’s model of the salutogenesis is based on his concept of the sense of coherence (SOC), which indicates whether or not inner and outer stimuli are comprehensive, manageable and meaningful to an individual [11]. These so-called generalized sources of resistance (comprehensibility, manageability and meaningfulness) develop, depending on individual stress and bonding experiences, as well as experiences regarding self-esteem, which were gained in childhood [12]. A strong SOC usually leads to an improved health status, whereas a low sense of coherence leads to health problems. Studies concerning this topic have shown that people with high levels of coherence are personally more convinced to handle stressful situations through existing resources than individuals with a low SOC [13].

Regarding the human-dog relationship, the influence of bonding experiences is of special interest, since human–animal bonds might also affect the strength of the SOC. This assumption is also supported by theoretical suggestions of Lars and Eileen Hegedusch, which suggest that the strength of the SOC is a potential measure of the positive effects of animals on humans’ wellbeing [14].

Since there is evidence that the relationship with a dog positively impacts the strength of the SOC and a strong SOC leads to a less stressful assessment of normative and non-normative stressors, it is likely that the human-dog bond and the assessment of stressful life events are also related. This assumption led to the research questions regarding to what extent the relationship with a dog influences the strength of the SOC, as well as the assessment of stressful situations. This would mean that dog owners, in the present study, are expected to show a higher SOC-value than non-dog owners, including a positive effect on the assessment of stressful life events.

Based on these considerations, the present study tried to examine if dog owners and non-dog owners differ in the strength of their SOC and in their assessment of stressful situations. An additional objective was to determine the actual impact of the human-dog bond regarding the strength of the SOC, as well as the assessment of stressful life events using correlations.

In order to verify these research questions, dog owners and non-dog owners were surveyed online and compared regarding the underlying theoretical constructs. To evaluate all possible distinctive features concerning the assessment of stressful life events, the items of the Social Readjustment Rating Scale (SRRS) were classified into normative and non-normative stressors. This categorization, described by Yoav Lavee, Hamilton I. McCubbin and David H. Olson [15] emphasizes that stress responses differ depending on the type of life event individuals are confronted with. Normative stressors are mostly predictable and expected and include challenges that most people encounter (e.g., giving birth, marriage or retirement). In contrast, non-normative stressors include unexpected life events which are not typical across life cycle (e.g., sudden death of a spouse or serious illness). Because of the unpredictable nature of non-normative stressors, they are, generally, presumed to be more difficult to cope with [15].

In order to evaluate the assessment of critical life events in a more differentiated way, additionally, the transactional model of stress according to Richard Lazarus was used. Based on his theory that the primary appraisal of stressors takes place in three gradations, the items of the SRRS were also classified as “challenge”, “threat” and “loss” [16]. Moreover, dog owners made subjective disclosures concerning the strength of the relationship with their dog.

Furthermore, additional calculations were performed on the theoretical basis that relationships and associated role-specific responsibilities have an impact on the assessment of stressful situations. Specific socio-demographic data (partnership, parenthood) were then further analyzed with regard to the variables of interest.

## 2. Materials and Methods

### 2.1. Study Design and Participants

An online survey was conducted in English and German, using a cross-sectional design. The survey test battery included open and closed questions regarding demographic data and standardized questionnaires concerning the human-dog bond (Lexington Attachment to Pet Scale), the sense of coherence (SOC-29-Scale) and the assessment of critical life events (Social Readjustment Rating Scale). Dog owners and non-dog owners were recruited using snowball sampling and surveyed online via different internet platforms. In total, 313 fully completed data sets were included in the statistical analysis (*n* = 313). The sample consists of 75.5% women, 23.9% men and 0.7% others, with an average age of 39 years (*M* = 38.55, *SD* = 15.106). Regarding marital status, the sample was mainly married (36.3%) or in a relationship (30.7%). Around a quarter of the participants stated to be single (24.8%), several participants were divorced (6.6%) or widowed (1.6%) at the time of the survey. Nearly one-third of the sample indicated that they had finished undergraduate or graduate school (bachelor’s or a master’s degree, 33.7%). A total of 14.7% stated that they had graduated high school and another 14.7% stated that they had graduated college. More than one-third of the participants indicated that they had attained another level of education (36.9%).

Regarding dog-ownership, more than half of the sample claimed to own at least one dog at the time of data receipt (52.8%). A total of 35.6% participants indicated that they had never owned a dog and 11.6% stated that they had owned a dog previously but did not at the time of the survey. Since the last group did not include a sufficient number of participants (n = 35), the data were not involved in the statistical analysis.

### 2.2. Instruments

#### 2.2.1. Demographic Data

Collected data included the present status of dog-ownership (current dog owners, earlier dog owners or non-dog owners), sex, age, highest educational attainment, job, current place of residence, nationality, marital status, number of children and ownership of another animal. These data could be important as possible influencing factors on the human-dog bond, the sense of coherence and the assessment of critical life events.

#### 2.2.2. Lexington Attachment to Pets Scale (LAPS)

The Lexington Attachment to Pets Scale [17] was used to measure the present experienced intensity of the bond between a dog and its owner. The questionnaire consists of 23 items that can be answered on a four-point rating scale (from totally disagree to totally agree). The LAPS commands a good internal consistency (Cronbach alpha of α = 0.928) and assesses the level of emotional attachment to cats and dogs. As all items are formulated in the present tense, only current dog owners were asked to answer this questionnaire.

Example statement: “I feel that my dog is part of my family”

#### 2.2.3. Sense of Coherence Scale (SOC-29-Scale)

Aaron Antonovsky’s Sense of Coherence Scale was used, for measuring the strength of the sense of coherence. It consists of 29 items, including the subscales “comprehensibility”, “manageability” and “meaningfulness” [18]. The response alternatives have a range between 1 point and 7 points and the internal consistency amounts to α = 0.92.

Example question: “Do you have the feeling that you don’t really care about what goes on around you?”

#### 2.2.4. Social Readjustment Rating Scale (SRRS)

The SRRS was used to measure the subjective experienced stress in relation to critical life events. It was developed in 1967 by Thomas Holmes and Richard Rahe and consists of 43 items that represent life events which entail a reorientation [19]. These items can be rated on a scale between 0 (no stress) and 100 (enormous stress).

Example items: “marriage” (normative stressor or challenge); “death of a spouse” (non-normative stressor or loss).

In order to evaluate all relevant characteristics of stressors, the items of the SRRS were divided into normative and non-normative stressors, as well as into “challenges”, “threats” and “losses”, on the basis of the theoretical principles of Yoav Lavee, Hamilton I. McCubbin and David H. Olson [15] and Richard S. Lazarus and Edward M. Opton [16], respectively.

#### 2.2.5. Subjective Assessment of the Human-Dog Relationship

In addition to standardised questionnaires, open questions regarding dog ownership were included in the test-battery. Dog owners were asked about their personal opinion regarding the subjective estimation of how their dogs affect them in stressful situations. The subjective viewpoints were compared with the measured findings.

### 2.3. Statistical Analysis

Analyses were computed with SPSS 24.0 (SPSS Inc., Chicago, IL, USA). Correlation calculations were conducted for measuring the mutual relations between the variables “human-dog bond” (determined by LAPS), “sense of coherence” (determined by SOC-29- Scale) and “assessment of critical life events” (determined by SRRS). Kolmogorov–Smirnov tests were implemented for the verification of the normal distribution. Differences between the groups “current dog owners” and “non-dog owners” were calculated using t-tests and Cohens d as an effect size measurement. To test the required conditions for t-tests, Kolmogorov–Smirnov tests of normal distribution and Levene’s Test of equality of variances were conducted prior to the examination. For all analysis, the significance level was set at *p* ≤ 0.05. After all the calculations were conducted, a correction of the *p*-values (method: Bonferroni correction) was executed and potential limitations were analyzed.

Since there is evidence that role-specific responsibilities affect the assessment of stressful situations and the results indicated an influence of the relationship with a dog on the assessment of stressful events, further calculations were conducted to investigate the general impact of relationships. For this purpose, demographic data regarding the marital status (married, single, domestic partnership, divorced or widowed) and current status of parenthood (children or no children) were included in the analysis. The family status was reduced to two variables (partnership and no partnership) and corresponding mean values regarding the assessment of stressful life events were compared. Additionally, general information regarding animal companionship (no animals, dogs, dogs and other animals) were used to calculate interactions between inner-species relationships (marital status and parenthood) and cross-species relationships (animal companionship) concerning normative stressors.

## 3. Results

### 3.1. Owners of A Companion Dog and SOC

In order to test whether the relationship with a dog influences the strength of the SOC (SOC-29 Scale), dog owners and non-dog owners were compared via t-test analysis. Table 1 represents the results of the calculation.

T-test analysis shows that there is no difference between dog owners and non-dog owners regarding the strength of the SOC.

### 3.2. Dog Companionship and the Assessment of Stressful Life Events

In order to verify whether the relationship with a dog influences the assessment of stressful situations, dog owners and non-dog owners were compared concerning the assessment of normative stressors and non-normative stressors, as well as the assessment of the categories “challenges”, “threats” and “losses” via t-tests. Table 2 and Table 3 represent the most important results of these calculations.

Data shows that there is a significant difference between dog owners and non-dog owners concerning the assessment of normative stressors. Moreover, data show no significant differences between dog owners and non-dog owners concerning the assessment of non-normative stressors. These results demonstrate that dog owners assessed normative stressors to be more stressful than non-dog owners.

Data shows a significant difference between dog owners and non-dog owners regarding the assessment of challenges. In the case of assessment of threats, data show no significant differences between dog owners and non-dog owners. Moreover, data show no significant differences between dog owners and non-dog owners concerning the assessment of losses. These results demonstrate that dog owners assessed daily hassles, but not threats and losses, to be more stressful than non-dog owners.

### 3.3. Impact of the Strength of the Human-Dog Bond on the SOC and the Assessment of Stressful Life Events

In order to determine the actual impact of the relationship with a dog on the strength of the SOC, as well as on the assessment of stressful situations (SRRS), Pearson correlations between the variables “human-dog bond” (LAPS) and “strength of the sense of coherence” (SOC-29-Scale) were conducted. Table 4 represents the most important results of these calculations.

Results show no significant correlation between the variables “human-dog bond” and “strength of the sense of coherence”. Moreover, data shows a small but significant positive correlation between the variables “human-dog bond” and “assessment of critical life events” to the effect that the stronger the relationship is rated, the more stressful the life events are assessed as being. Additional correction using Bonferroni, underlined the necessary and previously mentioned cautious interpretation of the significant results with a focus on the effect size calculation. Three of the six significant results stayed on a significant level of *p* ≤ 0.05 after the correction (mean differences between participants with partnership and participants without partnership regarding the assessment of non-normative stressors, mean differences between participants with partnership and participants without partnership regarding the assessment of losses, mean differences between participants with children and participants without children).

### 3.4. Inner-Species and Cross-Species Relationships and the Assessment of Stressful Life Events

To investigate the impact of inner-species relationships on the assessment of stressful life events, the respondents’ current family status, as well as their current parenthood status, were included in the analysis. The following charts (Figure 1 and Figure 2) demonstrate the most important mean differences.

Data shows that individuals with inner-species relationships tend to assess critical life events as being more stressful than individuals without stated inner-species relationships. Regarding the family status, data demonstrate that participants living in a partnership (*M* = 40.54, *SD* = 16.32) tend to rate normative stressors as being more stressful than participants who live alone (*M* = 38.08, *SD* = 13.69). Considering the differences concerning non-normative stressors, individuals in a current partnership (*M* = 68.90, *SD* = 13.40) assessed critical life events as being significantly more stressful (*t*(294) = 1.26, *p* = 0.002, *d* = 0.387) than individuals without a current partnership (*M* = 63.33, *SD* = 15.75). These data are in accordance with the results of the experienced stress regarding losses. Participants living in a partnership (*M* = 80.56, *SD* = 12.05) rated these items as being significantly more stressful (*t*(294) = 3.88, *p ≤* 0.001, *d* = 0.48) than participants who live alone (*M* = 74.01, *SD* = 16.13). Concerning parenthood and the assessment of critical life events, data also show significant differences between individuals who indicated that they had children and individuals who indicated that they do not have children. Concerning normative stressors, participants with children (*M* = 41.37, *SD* = 14.43) rated items as being more stressful than participants without children (*M* = 37.70, *SD* = 16.24). Data show a similar result regarding the assessment of non-normative stressors, which demonstrate that individuals with children (*M* = 68.67, *SD* = 14.62) rated these items as being more stressful than individuals without children (*M* = 65.18, *SD* = 13.91). Analogous results have been found concerning the assessment of challenges, threats and losses. Regarding the assessment of losses, data even show that participants with children (*M* = 80.65, *SD* = 13.38) rated these items as being significantly more stressful (*t*(294) = 3.08, *p* = 0.002, *d* = 0.359) than participants without children (*M* = 75.73, *SD* = 13.89).

Additionally, interactions between inner-species relationships and animal companionship in relation to normative stressors were tested. Therefore, five different categories regarding inner- and cross-species relationships were created (“Children yes/no”, “Partnership yes/no”, “No animals”, “Dogs”, “Dogs and other animals”). Figure 3 demonstrates the most important interactions.

Overall, these data indicate that individuals tend to rate normative stressors as being more stressful the more relationships they enter. The only exception are individuals living in a relationship without animals, who tend to rate normative stressors less stressful than individuals living alone without animals.

### 3.5. Subjective Assessment of the Human-Dog Relationship

Results of the subjective assessment of the human-dog relationship show that most dog owners experience their dogs as helpful when coping with stressful situations. Statements like “My dog notices when I’m not feeling well. He then behaves very considerately” and “My dog already helped me through very stressful times. He gave me the love I needed in that time”, suggesting that dog owners attribute their dogs as having a helpful role during stressful situations.

## 4. Discussion

Based on the findings of the presented dataset, it might be assumed that an overestimation of the dog’s protective role regarding stress has taken place in public media and research. However, the results draw another picture of the dogs’ role to the often-described protective factor in stressful situations. The presented calculations show no evidence of the influence of dogs regarding the strength of the sense of coherence, which contains the resources available to cope with stress. This leads to the interpretation that there is no protective factor detectable in the current dataset. The results of this study are in contrast to earlier stated assumptions, indicated by Eileen and Lars Hegedusch, who stated that dogs at least influence the SOC indirectly [14].

Furthermore, it appears that dog owner- and companionship causes additional stress when coping with critical life events. This assumption is based on the results regarding dog ownership as well as the human-dog bond in connection to the assessment of critical life events. Data show a significant difference between dog owners and non-dog owners in regard to the assessment of normative stressors and challenges, which might suggest that dog-ownership particularly affects the stress experienced in everyday life. Daily hassles are experienced as being more stressful when living with a companion dog. Moreover, a significant positive correlation between the strength of the human-dog bond and the assessment of stressful life events shows that the stronger the relationship between a dog and its owner is perceived as being, the more stressful critical life events are rated. This result marks another difference to previous findings, which emphasize the positive influence of the relationship with a dog regarding the subjective perception of stress [20]. One potential explanation for this phenomenon is the responsibility owners assume for their dogs, which modifies the relevance of critical life events. Animals convey a feeling of being needed, which is why they often develop an increased sense of responsibility towards them [21]. The associated caregiving might be an explanation for the fact that the assessment of critical life events increases in proportion to the strength of the human-dog bond.

In this context, previous studies underline that, especially, role-specific responsibilities affect the subjective experience of stressful situations [21]. The more responsibilities individuals undertake by entering deep relationships, the greater the range of reactions to critical life events [22]. This facet can also be found within the present study, showing that the assessment of critical life events tends to get more stressful the higher the number of relationships (inner- and cross-species) individuals report. In regards to this, data shows that participants of the current study who indicated that they live in a partnership, to have children, to own a dog or to have other animals tended to assess critical life events as being the most stressful. Moreover, calculations regarding interactions between inner- and cross-species relationships demonstrate that individuals living alone without animals tend to rate normative stressors as being less stressful than individuals living in a relationship without animals. This result reinforces the theory that particularly, cross-species relationships cause additional stress.

The fact that owning an animal (especially dogs) plays a relevant role regarding the assessment of stressors once more demonstrates the highly relevant status of animals within the family system. This aspect can also be found in current studies, which find that including pets in a key social ingroup (i.e., family) improves wellbeing [2]. Although this result seems to contradict the interpretations of the present study, it rather underlines the changing role of animals into important family members. On the basis of this assumption and the fact that the mentioned study does not refer to stressful situations, it might be assumed that the change within cross-species relationships affects human experiences and behavior in many ways.

Specific findings furthermore demonstrate that besides the relationship with a dog, factors such as current parenthood and partnership situations, especially, lead to significant increases regarding the assessment of critical life events. According to John Bowlby’s and Mary Ainsworth’s bonding theory, these results lead to the assumption that the human-dog bond not only changed during recent years but is also comparable to the bond between parents and their children [23]. This aspect is also confirmed by several studies focusing on the consequences of critical life events. A study from 1989 examined the psychological impact of the Three Mile Island Incident—a nuclear meltdown which happened in 1979 in Pennsylvania and was rated as an accident with wider consequences [24]. The results of this study show that, in comparison with other local residents, mothers experienced elevated levels of stress and psychological symptoms like anxiety and depression. The authors concluded that one possible explanation for this phenomenon could be the parental role involvement because the perceived harm to a child’s health is contained in the most important appraisal associated with distress [24]. We assume that the role of dogs has changed over the past years from hunting partner to companion and family member, which might explain why current dog owners in this study assessed critical life events as being more stressful than non-dog owners.

Although the results of the current data set indicate that previous statements regarding the protective function of dog ownership might have been overestimated, it must be stated that personal subjective viewpoints concerning similar aspects often differ from measurable findings in general. Individual statements of the subjective assessment of the human-dog relationship indicate that dog owners assign a helpful role to their dogs during stressful situations. This aspect leads to the interpretation that there might be a protective effect within dog ownership which could not be determined using the presented theoretical constructs, measurements and/or calculations. This seems to be one critical aspect of the present study which needs to be considered in further research.

However, findings show that the strength of the human-dog bond affects the assessment of critical life events. It can be considered as a statistical tendency that the more inner- and cross-species relationships individuals cultivate and the more other individuals (increasing with the number of different species) they consider as part of their family, the more stressful critical life events are experienced as being.

Finally, it must be noted that, especially, in the emerging field of human–animal interactions, it is highly relevant to point out positive, as well as critical, aspects and present significant, as well as non-significant, results. Publication bias needs to be minimized. The present results might offer an opportunity to underline this relevant perspective in current research.

## 5. Limitations

Despite several statistically significant results, it is important to mention that the relevance of the results in the present study needs consideration. The effect sizes in all calculations are very small. That is why it must be reflected that the differences and correlations regarding the assessment of critical life events might be incidental, which is also evident from the results of the *p*-value correction. Considering the fact that this study did not focus on examining the motivations of dog ownership in depth, this might be seen as a limiting factor in the interpretation of the results. The present findings might also be a result of the fact that individuals who rate critical life events as being more stressful in general own a dog, as opposed to individuals rating critical life events as being less stressful. Another critical aspect of the presented study is the sampling approach using voluntary participants. In addition to the fact that the general population of dog owners is unknown, which is a problem for representative sampling per se, participants were recruited online and were given the chance to voluntarily take part in the study or not. This clearly leads to a self-selected sample and, therefore, to a potentially biased result. However, it must be mentioned that, according to Internet World Stats, the internet penetration rate is quite high already (87.7% Europe, 89.9% North America) [25]. A potential solution for the future would need to include knowledge about the researched general population (all dog owners) to compare the results from a representative group with the presented findings.

## 6. Conclusions

The relevance of relationships with dogs seems to be even more important than previously expected. Most results in this study indicate that the human-dog bond impacts a dog-owners’ life and that the role of dogs has changed into the role of a family member. Regarding the sense of coherence, which was assumed as an important parameter within the present study, it might be argued that based on our findings, the SOC has less impact on the self-concept than expected. This aspect should be taken into consideration in further research regarding human–animal bonds.

Taking all results into account, the statistical tendency that dog ownership leads to additional experienced stress seems obvious. This leads to the assumption that an overestimation of the dog’s protective role regarding stress may have taken place. Although personal subjective viewpoints differ from measurable findings, assumptions in public media and research on dogs and their positive influence on wellbeing on humans play a role that should not be underestimated. Future research should examine this aspect and analyze the difference between inducted ideas regarding the benefits of animals and actual measurable profits.

## Figures and Tables

**Figure 1 ijerph-16-03664-f001:**
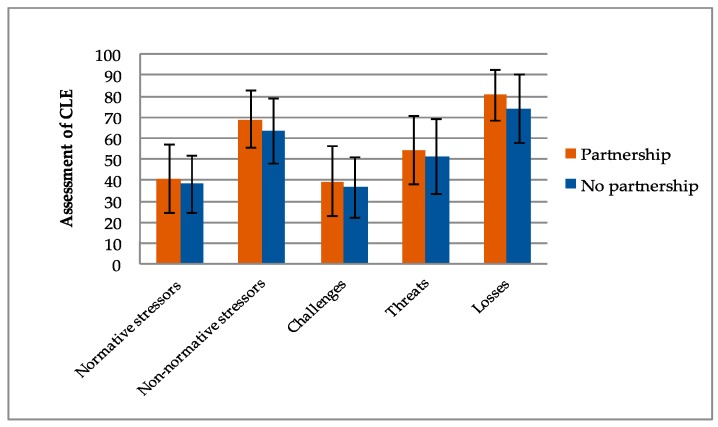
Mean differences between participants with partnership and participants without partnership regarding the assessment of normative stressors, non-normative stressors, challenges, threats and losses. Error bars represent standard deviation.

**Figure 2 ijerph-16-03664-f002:**
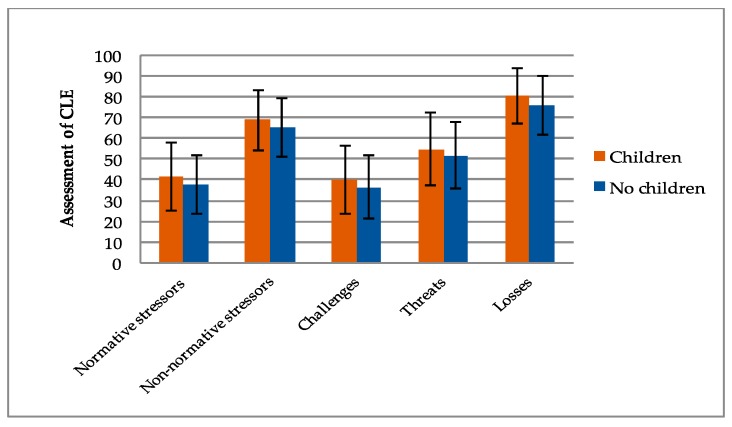
Mean differences between participants with children and participants without children regarding the assessment of normative stressors, non-normative stressors, challenges, threats and losses. Error bars represent standard deviation.

**Figure 3 ijerph-16-03664-f003:**
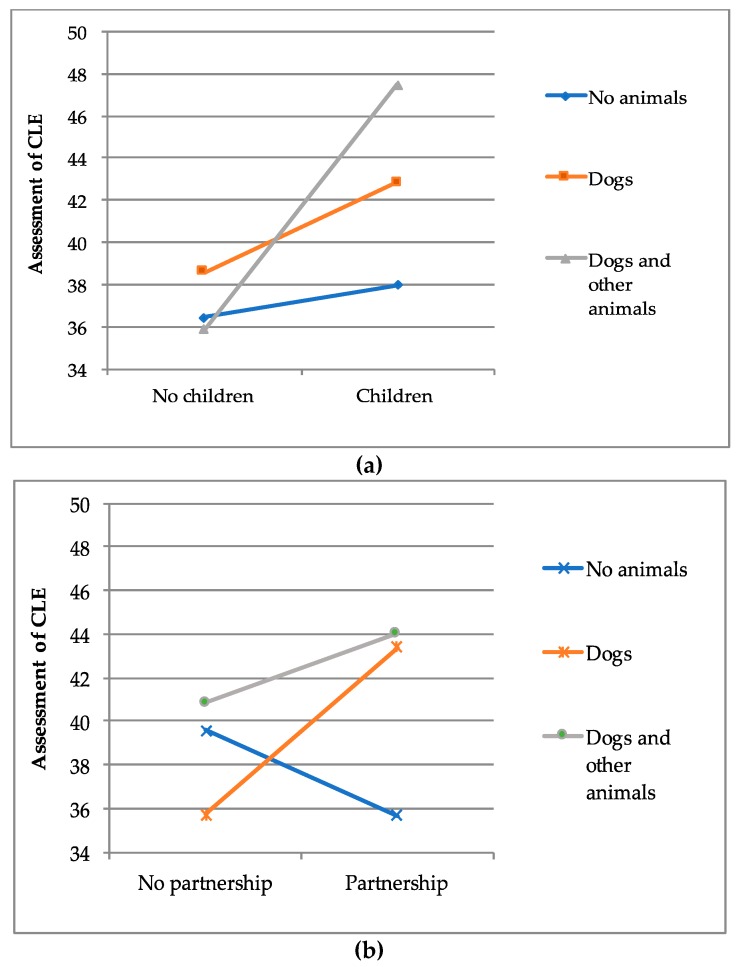
Interactions between inner- (**a**) and cross-species (**b**) relationships regarding the assessment of normative stressors.

**Table 1 ijerph-16-03664-t001:** Mean differences of the groups “dog owners” and “non-dog owners” regarding the strength of the sense of coherence.

Sense of coherence	**Dog Owners**		**Non-Dog Owners**					**95% CI**	
	M	SD	M	SD	*d*	*t*(250)	*p*	LL	UL
	113.5	8.03	113.5	7.1	0	0.033	0.973		1.9

Note: CI = Confidence Interval. LL = Lower Limit. UL = Upper Limit.

**Table 2 ijerph-16-03664-t002:** Mean differences between the groups “dog owners” and “non-dog owners” regarding the assessment of normative and non-normative stressors.

normative stressors	**Dog Owners**		**Non-Dog Owners**					**95% CI**	
	M	SD	M	SD	*d*	*t*(265)	*p*	LL	UL
	41.85	15.97	37.51	15.18	0.277	2.22	0.027	0.49	8.18
non-normative stressors	**Dog Owners**		**Non-Dog Owners**					**95% CI**	
	M	SD	M	SD	*d*	*t*(265)	*p*	LL	UL
	68.34	13.97	66.45	14.91	0.132	1.06	0.290	−1.62	5.4

Note: CI = Confidence Interval. LL = Lower Limit. UL = Upper Limit.

**Table 3 ijerph-16-03664-t003:** Mean differences between the groups “dog owners” and “non-dog owners” regarding the assessment of challenges, threats and losses.

challenges	**Dog Owners**		**Non-Dog Owners**					**95% CI**	
	**M**	**SD**	**M**	**SD**	***d***	***t*(265)**	***p***	**LL**	**UL**
	40.62	16.33	36.04	15.52	0.286	2.3	0.022	0.651	8.52
threats	**Dog Owners**		**Non-Dog Owners**					**95% CI**	
	**M**	**SD**	**M**	**SD**	***d***	***t*(265)**	***p***	**LL**	**UL**
	54.79	16.49	52.61	17.04	0.13	1.04	0.29	−1.93	6.27
losses	**Dog Owners**		**Non-Dog Owners**					**95% CI**	
	**M**	**SD**	**M**	**SD**	***d***	***t*(265)**	***p***	**LL**	**UL**
	79.45	13.37	77.67	14.2	0.13	1.04	0.29	−1.59	5.15

Note: CI = Confidence Interval. LL = Lower Limit. UL = Upper Limit.

**Table 4 ijerph-16-03664-t004:** Correlation between the human-dog bond and the strength of the sense of coherence, as well as between the human-dog bond and the assessment of critical life events.

		SoC	Assessment CLE
Human-dog bond	*r*	0.13	0.2
	*p*	0.109	0.013

Note: SoC = Sense of Coherence. CLE = Critical Life Events.

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
