# Peer review of "And in the Middle of My Chaos There Was You?—Dog Companionship and Its Impact on the Assessment of Stressful Situations"

_ijerph, 2019, doi:10.3390/ijerph16193664_

Round 1

Reviewer 1 Report

The authors note that animal companionship has been found to lower stress level in humans and suggest that Antonovsky’s theory of solutogenesis may be relevant for theoretical explanations. They include measures of two Antonovsky related constructs (Sense of Coherence and Critical Life Events) in an internet survey, which also contains a measure of pet bonding (the Lexington Attachment to Pets Scale) and some demographic questions, and they report data from a convenience sample of 313 respondents.

They find that in their sample, dog owners are not less impacted by Critical Life Events than non-dog owners. Subsets of this measure may suggest the contrary. They find no difference between dog owners and non-dog owners on Sense of Coherence and no correlation between pet bonding and either of the Antonowky related measures. They interpret this as evidence against a stress buffering effect of companion animals. I find that this might be true but there is no convincing argument for this rather than other interpretations.

Generally, I find that the data may be interesting, and testing explanatory potentials for the area of a theoretical account that appeals to many practitioners, certainly is. However, in this manuscript, the flow of the argument is far from clear, and the use of the data confused me a bit. For example, no explicit hypotheses are stated, results are presented as separate analyses of the same data, the majority of the results section report on analyses that seem unrelated to the rationale given in the introduction, and the discussion seems to go in various other directions.

I think the manuscript could work if it becomes a shorter paper with one tight argument.

The discussion includes a reference to other authors who suggested that Antonovsky’s work has explanatory value regarding positive impact of companion animals for health (258-60). This is not mentioned in the introduction, whereas using it as the starting point for the manuscript might produce the research question you need. So might other solutions. My main concern is that the manuscript needs a clear research question that is argued for in the introduction and lead actor in the results and conclusion section.

I therefore suggest rewriting of the manuscript. Given that, most specific/minor comments may be irrelevant at this point. However, these considerations may still be useful:

- When using non-English references: If the original was actually in English, I think it is preferable to refer to that one. If not, translating the title [in brackets] is useful.

- When you refer to the work of other authors, please consider adjusting the phrasing to more clearly distinguish between findings, theoretical suggestions, postulates etc.

- When performing multiple analyses on the same data set, consider adjusting the p-value.

- You provide several tables with just one result each. I think some several of them could be combined and would become more useful if they were.

- You mention effect sizes but they are not in the tables. They could usefully be, in general. And in the case of correlations, it seems strange to me that the tables apparently report the significance level found instead of the actual correlation coefficient (i.e. the effect size) that is typically reported in correlation tables.   

- The literature on the human-animal-bond is growing almost exponentially now and you may be able to find more recent references (including reviews) for some of your assertions.

- I think that the text bits about checking for distribution of data before choice of test (lines 180-182, 18-186) might be better placed in the method section.

- In the results section, often information in tables and text are redundant. I suggest giving it only once.

- When reading results in text, I find it more helpful to be told the direction of a difference than just the existence of one. For example, in line 190-191 I would write, “Concerning normative stressors, dog owners assessed these to be higher than non-dog owners did”. (and cf. my last point, the actual figures would go either here or in the table, not both).

- I missed a section/paragraph on limitations of your study, including issues such as self-selected samples.

- Although I like the title, I don’t think it matches the findings. At least it requires a “?” after the “you”.

Reviewer 2 Report

This is an interesting manuscript that provides evidence for dog ownership being associated with WORSE stress response outcomes, which is unusual and worth publishing. However, I do have some concerns about the present state of the ms, that I would like the authors to address.

The abstract is very unclear. It should be rewritten with less jargon and more description. The authors should not expect the reader to understand Antonovksy's sense of coherence, or what a 'normative' vs 'non-normative' stressor is. These things can be briefly described, which will make the entire abstract clearer. Also, the authors state in the abstract that there was a sig diff between owners and non-owners regarding the assessment of normative stressors, but they don't say the direction - it would be helpful if they wrote that owners report lower levels of stress in those circumstances than dog owners. That's much more instructive than just saying there is a difference.

L33 - pet? or working animal?

L42 - assume, or report?

L58 is a sentence fragment

L73-75 - the sentence structure needs work. It's unclear as is.

L77 - what is 'overload/underload' of stress? Please explain

L81-84 - this argument is a stretch to me. Either find more citations, or just drop it altogether.

Participants - since 'former dog owners' were not included in analysis, how many were retained?

L117 - resistance should be residence?

L118 - material should be marital?

L122 - sentence fragment.

Please provide an example item of the LAPS, as well as the SOC and the SRRS.

L140-150 - is this justified based on previous research? Or is it just something the authors thought would make sense given their question? Also - this entire para would be better in the intro. Adding theoretical justifications in the methods that haven't been mentioned in the intro is confusing. The authors should discuss all of these theoretical considerations in the intro. Currently, it's unclear what 'challenge', 'threat', and 'loss' really mean, because they haven't been sufficiently explained.

Results

Section 3.1- dog-companionship would be better as 'owners of a companion dog'

L160-167 should go into the analysis section.

I don't think the authors need table 1. The info is already presented in the text anyway, so the table just wastes space. Same with all instances where the t-test results are presented in text as well as in the table. Delete them from the text, and just leave the interpreting statement instead.

Table 2 - is the .109 the p-value or r-value? This is made clear in text but the tables and figures should stand alone. Just specify that this is the p-value, or just drop the table altogether, since the info is already reported in text.

Same with the other tables in this section.

L204 - but the effect size is small, which is worth mentioning.

Fig 3 - only children are presented. Could the authors have another table, sitting side by side with the current one, for partnership?

L297 - lust years?

L312-319 - this belongs in the results section. New results should not be presented in the Discussion.

L345 - inducted imagination? What does that mean?

L346-351 would be better in the discussion.

Round 2

Reviewer 1 Report

Thank you for the opportunity to read revised version of this interesting study. I find that the study has merit and the manuscript has been improved. Some of my original concerns have been dealt with but some not quite, and I now also add a list of minor suggestions that seemed premature for the original version. Some are about language but I am not English myself and may have missed some language problems. I think the manuscript would profit from thorough checking by a native English speaker.

Issues:

Section 3.4.: This entire section is not related to the theoretical argument or RQ given in the introduction. That gives the impressions of a last-minute-save or wandering off. I think the manuscript would be much improved by either skipping this section entirely, or arguing coherently why it is relevant to the theoretical argument.

Section 3.5.: This section seems to leave us with a contradiction between what people report when directly asked (the dog reduces stress), and what the measures used find (rather the the contrary). I think that you need to address this in the discussion or skip section 3.5.

In the discussion (section 4) I missed this reflection anyway. Theories and measures may not work outside their original domain, and maybe what you found is that this theory does not capture the HAB domain well (or the measures by which you operationalised the theory, do not). That discussion would add to the value of the article.

In my review of the original article, I suggested that it should be shorter. You have prolonged it instead. However, I think that most of the additions are relevant, so I concede this point.

Minor issues:

21: ”focusing on SOC” – unclear phrasing, what focuses on SOC?

23: ”was” should be ”were”

25: ”In this context” might be replaced by ”Contrary to expectations” to clarify

37: ”which” – refers to?

41: ”could not only be determined” sounds a bit strange to me. Do you mean ”are not only found” or ”could be expected not only…” ?

43-44: ”long term studies indicate” – plural phrasing but one reference

45-46: please check phrasing

48: I think a stronger reference than [6] is warranted – I suggest using a review or a least a more recent study (preferably in English)

52: ”The authors’” = who? You, or Herzog but a misplaced ’ ?

61: ”draw several strategies from” – sounds a bit odd to me, is it correct English?

66: ”successful” should be deleted

66-67: ”The so-called salutogenetic concept is based on the sense of coherence (SOC) which indicates” – should this be: Antonovsky’s salutogenetic concept is based on his concept sense of coherence (SOC), which indicates

74: ”convinced to” – consider rephasing

78: ”demonstrate” – how do they demonstrate that? Do you mean suggest?

81-84 seems to be in contradiction. L81:Since there is evidence that the relationship to a dog impacts the strength of the SOC positively”. L84:”…led to the research questions whether the relationship to a dog influences the strength of the SOC and…” How can evidence that x exists lead to the research question does x exist?

86: ”study show” – for clarity, I suggest inserting ”is expected to” between ”study” and ”show”

92: ”On order to verify these research questions” – consider rephrasing

94: ”SRRS” = ?

100: typically – grammar?

101: ”apply to”- correct word?

102: ”according to” - correct phrasing?

104: ”gradations” - correct word?

105-108: What is described here? Is this a summary of the results of your study, and if so, is this the place to put it? If it is a summary of another study, please explain and provide reference.

121: ”The surveyed population” – do you mean sample?

126: ”before” – word should probably be deleted

140+148: Are the Chronbach α levels the levels found for these tests in your sample? If not, I suggest adding the ones that you found in your sample in the method or results sections. Unless tests are robust, α will vary between samples, and low α reduces your chance of statistical significance when using the measure, so this could be relevant for interpretation of your findings.

178-187: This section goes some way to explain the addition of an analysis that is not part of the theoretical argument or RQ in the introduction but I don’t think the method section is the place to put it.

198: ”verify” – is this the right word?

Table 2: Please remove duplicate headlines

Table 3: Please remove duplicate headlines

Between table 2 and table 3: Please clarify the relation between the two categorizations in table 2 and table 3, respectively. Are table 3’s categories subsets of normative stressors, of non-normative stressors, or across normative and non-normative stressors?

217: For clarity, I suggest adding ”, but not threats and losses,” between ”hassles” and ”to”

Table 4: I don’t think this is a standard way of reporting correlations. If you consider three varibles (here: HAB, SOC, CLE) articles typically report correlations between all three in a matrix and just add a * or emphasis to those that are statistically significant.

292: ”as” – I think this should be ”than”

327: ”postulate” – this refers to empirical studies so I think ”find” is the relevant word

333-347: This paragraph reads like a digression and I suggest that you delete it

391: “has taken place” – I suspect that you are right, but given the limitations of the study (and, indeed, any single study), I think that the more appropriate phrasing would “may have taken place”.
